# A realist perspective on optimizing community health workers' roles and functions to deliver integrated people-centred care

Usangiphile E. Buthelezi[1,2*], André J. van Rensburg[1], Mosa Moshabela[2,3], Sanah Bucibo[1], Noxolisa Radebe[1], Zamasomi Luvuno[1], Tasneem Kathree[2], Arvin Bhana[4,5], Inge Petersen[1,6]

**1** Centre for Research in Health Systems, University of KwaZulu Natal, Durban, South Africa, **2** School of Nursing and Public Health, University of KwaZulu-Natal, Durban, South Africa, **3** University of Cape Town, Cape Town, South Africa, **4** Centre for Research in Health Systems, School of Nursing and Public Health, UKZN, Durban, South Africa, **5** Health Systems Research Unit, South African Medical Research Council, Durban, South Africa, **6** Institute of Global Health, University College London, London, United Kingdom

* sahbut1@gmail.com

## Abstract

Community Health Workers (CHWs) play a crucial role to support health care delivery in underserved communities. Although the value of CHWs' contributions is widely recognised, there is limited evidence on the mechanisms that enable CHWs to deliver people-centred care. Using a realist evaluation approach guided by WHO's Integrated People-Centred Health Services (IPCHS) framework, the study focused on how different contexts and mechanisms interact to align with the IPCHS strategies to shape CHWs' capacity to deliver people-centred care. This realist qualitative study was conducted in five rural communities in KwaZulu-Natal, South Africa. Data was collected through structured observations of CHWs' interactions with households; interviews with CHWs, service users, household decision makers, outreach team leaders (CHWs' supervisors), and clinic operational managers. Data was further corroborated through three focus group discussions with CHWs. Using thematic analysis and realist evaluation methods, we identified Context-Mechanism-Outcome (CMO) configurations influencing CHWs' delivery of people-centred care, followed by refinement of the programme theory and development of middle-range theories. The study identified meta-mechanisms (trust, legitimacy, and motivation) that operate across all domains of the IPCHS framework and underpin the ability of CHWs to engage communities, coordinate care, and deliver integrated, people-centred services. These meta-mechanisms are triggered within enabling conditions, notably formalized supervision, CHW integration into the formal health system, and intersectoral collaboration. However, governance gaps such as precarious employment, inadequate remuneration, poor resourcing, lack of data feedback loops, and insufficient institutional recognition of CHWs' intersectoral role undermines these interactions, resulting in the poor

**Data availability statement:** The dataset extracted and used for this study is available in NVivo format at the following link: https://figshare.com/articles/dataset/A_Realist_Perspective_on_Optimizing_Community_Health_Workers_roles_and_functions_to_deliver_Integrated_people-centred_care/29424230. The availability of the data share is in line with FAIRSharing principles (https://fairsharing.org/) and Wellcome Open Research Data Guidelines (https://wellcomeopenresearch.org/for-authors/data-guidelines). For more information on the data set contact crh@ukzn.ac.za.

**Funding:** This research has been supported by funding from the UK Foreign, Commonwealth & Development Office (FCDO), Medical Research Council (MRC) and Wellcome Trust (MR/V015044/1 to IP). The work reported herein was supported by the South African Medical Research Council through its Division of Research Capacity Development under the SAMRC Researcher Development Award with funding received from the South African National Department of Health (UB). The funders had no role in study design, data collection and analysis, decision to publish, or preparation of the manuscript.

**Competing interests:** The authors have declared that no competing interests exist.

delivery of IPCHS. The study contributes to policy discussions by providing middle-range theories that explain how, why, and when CHW-led people-centred interventions fail or succeed. Critical findings include the need for a dynamic Integrated, Mechanism-Sensitive Model of the IPCHS and governance reforms that include structured workforce integration for adequate resourcing and intersectoral action.

## 1. Introduction

Community Health Workers (CHWs) contribute significantly in expanding access to healthcare services in underserved communities [1,2]. They serve as the first point of contact with the health system for communities, providing important health services including health education, disease prevention, home-based care, and referrals to the formal healthcare system [3–7]. In South Africa, CHWs form part of the Ward-Based Primary Healthcare Outreach Teams (WBPHCOTs) as part of the country's Primary Healthcare (PHC) re-engineering initiative. These teams bring healthcare services to communities, especially in rural and underserved regions, liaising between communities and healthcare systems. Each WBPHCOT is made up of six to ten CHWs, and is led by an Outreach Team Leader (OTL), usually an enrolled nurse responsible for CHWs' supervision, oversight, and data capturing [8]. The team approach is meant to improve service delivery at the community level and to strengthen the coordination and linkage between community-based care provided by WBPHCOTs and formal health care services within the health system. A number of studies have shown that CHWs improve maternal and child health outcomes [6,9], enhance adherence to chronic medication [10,11], and reduce disease burdens by delivering decentralized healthcare services [12]. Furthermore, CHW-led interventions have been instrumental in addressing social determinants of health by integrating health services with social support systems, hence improving overall well-being [13,14].

To reach Universal Health Coverage (UHC) and Sustainable Development Goal (SDG) 3, good health and well-being, governments and international health organizations, including the World Health Organization (WHO), have called for the inclusion of CHWs as part of primary healthcare [15]. The WHO's Integrated People-Centred Health Services (IPCHS) framework further emphasizes the need for CHW programmes to engage communities, reorient the model of care, create an enabling environment, and coordinate services across sectors to achieve accessible, responsive and people-centred care [16]. However, despite the importance of CHW roles, they often face systemic challenges that hinder their effectiveness including role confusion, insufficient resources, poor compensation, and governance challenges [17].

While there is substantial evidence on the impact of CHWs with regards to primary healthcare delivery [18], critical knowledge gaps persist. First, there is a dearth of evidence on mechanisms that enable CHWs to successfully navigate their complex roles in providing integrated, people-centred care. Existing studies have predominantly focused on CHW programme outcomes rather than contextual factors and mechanisms that influence their effectiveness [19]. Second, there is limited research

on how different governance structures, accountability mechanisms, and intersectoral collaboration may enhance CHW performance and the quality of services they deliver [20]. The majority of CHW programmes are often integrated into fragmented health systems with inconsistent policies for workforce integration, compensation and supervision, but limited empirical evidence exists about how these affect CHWs' ability to deliver people-centred care. Furthermore, studies to date are mostly descriptive or cross-sectional and do not consider dynamic, context-dependent nature and mechanisms that influence CHW outcomes. Consequently, there is a dearth of middle-range theories that elucidate how, why and in what circumstances CHW interventions succeed or fail. Therefore, due to the lack of evidence that explains the mechanisms of change, a realist evaluation approach was deemed appropriate for this study.

Grounded in the IPCHS framework, this study systematically investigates how different interactions between context, mechanisms, and outcomes (CMO) configurations influence CHW's roles and functions using a realist evaluation approach. This research explores the underlying processes that enable or hinder CHW interventions towards people-centred care in the real-world setting.

## 2. Materials and methods

### 2.1. Study design

This study was conducted between 01 November 2022–26 July 2023 in uMgungundlovu District, KwaZulu-Natal, South Africa. The study employed a case study approach and realist evaluation to examine the interactions between CHWs, Outreach Team Leaders (OTL), Clinic operational managers (for clinics that does not have OTLs) and household members (service user and decision maker) during home visits Using the IPCHS framework as a guiding framework, we adopted a realist evaluation study approach to critically examine and refine the roles and functions of CHWs in the provision of integrated and people-centred care [16,21]. Realist evaluation examines the effectiveness of programmes by asking which interventions succeed for which people in which situations and through what processes [21]. This realist evaluation was conducted as part of the overall study titled *"How to strengthen people-centred community health system competence in South Africa" (MRC Reference: MR/V015044/1)*, which aimed to investigate the ways in which the effectiveness and responsiveness of community health systems can be supported by enhancing the competence of the multiple stakeholders involved, including CHWs.

We adhered to the Realist and Meta-narrative Evidence Syntheses: Evolving Standards (RAMESES) II guidelines for study design and reporting [22]. The realist approach is outlined in Fig 1 [21].

### 2.2. Step 1: Initial programme theory (IPT) development

In general, programme theories explain how interventions produce intended or unintended outcomes, while offering plausible reasons why an intervention is expected to work or not work in a given context [24]. An IPT was developed guided by the IPCHS framework and expert consultations with a research team consisting of researchers with expertise across different fields including public health, nursing, psychology, sociology, and political science who have published a significant amount of work on primary healthcare, health systems strengthening and CHWs. The IPT was initially based on the WHO's IPCHS framework which identifies five inter-related cross-cutting strategies for people-centred care as shown in Table 1. These strategies offered a conceptual framework for understanding how, why and under what circumstances CHW interventions can deliver integrated people-centred care.

The IPT was further refined by a realist literature review to test, validate, or refute the IPT and identify associated CMO configurations to explain factors influencing the roles and functions of CHWs in people-centred health services. This process of refinement was iterative and systematic. In total the review synthesized 101 CMO configurations from 36 studies in 14 sub-Saharan African countries. These CMO configurations were subsequently condensed through an iterative process of refinement, including research team discussions, and synthesis meetings to 17 higher-order CMOs, which aligned with the five IPCHS strategies and were based on theoretical and empirical underpinnings (Fig 2). The key mechanisms

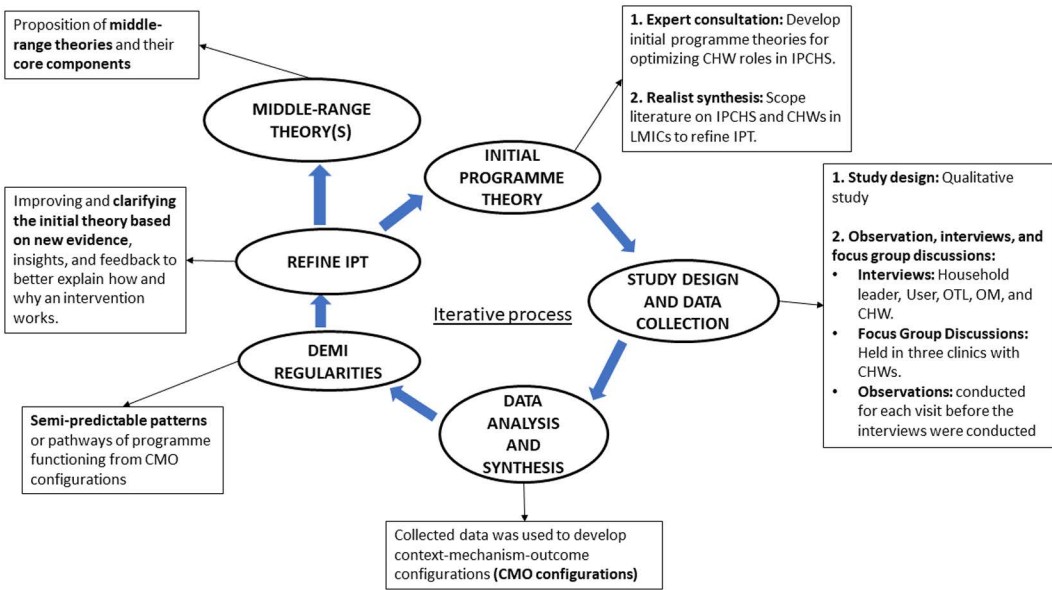

**Fig 1. Realist cycle of inquiry followed in the study [21,23,24].**

and outcomes included in the refined programme theory were trust-building, structured supervision, CHW integration into the formal health system, and intersectoral collaboration. The refined programme theory from the realist synthesis process, served as an initial programme theory for this study pending refinement through primary data. Details of the full IPT development process are available in a publication by Buthelezi et. al., 2025 [26].

### 2.3. Step 2: Data collection

**2.3.1. Sampling and participant recruitment.** *Interviews and Observations*: The data sources, methods of data collection, and their application in the analysis are summarised in Table 2. The study was conducted in five rural communities in KwaZulu-Natal, each aligned with a clinic catchment area. In each community, three households were purposively selected in consultation with a CHW, the Clinic Operations Manager (OM), and an OTL. During these consultations, the study team outlined the study objectives, emphasizing a focus on people-centred care for individuals with multiple chronic conditions. This focus was informed by evidence, including NICE guidelines, which recognize people-centred care as best practice for managing chronic multimorbidity [27]. These guidelines recommend that services prioritize what matters the most to patients by considering their preferences, life circumstances, and minimizing the treatment burden through holistic and person-centred planning. This aligns closely with the IPCHS framework used in our study and supports the focus on people-centred care. Using existing household profiles, CHWs identified families with a service user living with two or more chronic conditions who were scheduled for a CHW visit during the study period.

One structured observation and three interviews were conducted per household. A total of three households were visited per community, with the first and second visit focused on the structured observation and interviews with the attending CHW, the service user, and household decision maker. The rationale for interviewing the household decision-maker (HDM) was that they are responsible for making household decisions where the client resides. As such, they influence the patient's livelihood or well-being by providing key social determinants of health, such as food, water, and shelter. Where a single individual served as both the household leader and user, both interview questionnaires (user and HDM) were administered to ensure comprehensive data collection.

PLOS Global Public Health

**Table 1. WHO Integrated People-Centred Health Services Framework [25].**

| | |
|---|---|
| Engaging and Empowering people and communities | The initiative enables individuals to make informed health decisions while promoting community involvement in maintaining healthy environments and equipping informal caregivers with necessary education and support for better performance. The initiative concentrates on developing customized care through cooperative production to meet particular needs. |
| Strengthening Governance and Accountability | This strategy refers to the health system requirement to involve participatory governance which includes stakeholder involvement in developing policies and making decisions as well as evaluating performance throughout every organizational level from policymaking to clinical practice. |
| Reorienting the model of care | This strategy requires moving from inpatient treatment facilities to outpatient and ambulatory care services and shifting from curative methods towards preventive healthcare strategies. The approach places priority on funding holistic and comprehensive healthcare solutions that cover health promotion and preventive measures. |
| Coordinating services within and across sectors | This strategy requires health services to focus on people by connecting providers across different sectors and creating referral systems while linking health services with other sectors. This approach includes intersectoral collaboration at the community level to address social determinants of health. |
| Creating an enabling environment | This strategy entails that an enabling environment must first be established to effectively implement the above mentioned four strategies while uniting all stakeholders to drive transformational change. The establishment of an enabling environment requires multiple processes which encompass leadership and management changes alongside improved information systems, quality improvement methods and workforce reorientation. |

The HDM was identified by the service user after conducting structured observations, where they were directly asked, *"Who is the head of this household and the person who makes decisions?"* Furthermore, we also recognised that household decision-making is often distributed and may vary by domain - for instance, one person may control finances while another influences dietary or caregiving decisions. Since the study focused on people living with chronic conditions, we gave more weight to decision-making relevant to the user's health and care needs, including diet and medication adherence. If the nominated HDM was unavailable (which we never encountered in this study), the protocol was to reschedule to when feasible; otherwise, interview another adult familiar with household decisions and document accordingly.

The third visit had an additional interview with either the Outreach Team Leaders supervising CHWs or Clinic operational managers if there was no Outreach Team Leader allocated for that particular community. The purpose of the interviews with Outreach Team Leaders or Operational Managers was to gain supervisory and managerial perspectives on CHWs' roles, performance, integration into the formal health system, and the challenges they face in delivering people-centred care. The interviews focused on supervision structures, CHW accountability, resource availability, referral and feedback systems, and collaboration within the Department of Health. These insights were essential for triangulating findings from CHW and household interviews and understanding how institutional and governance contexts influenced CHW performance. Each household visit was considered a single case. In total there were 15 households included in the study resulting in 15 cases (S1 File). To supplement the findings from the interviews and observations of CHWs, focus group discussions (FGDs) were conducted with CHWs in three of the five clinics to corroborate the findings. These interviews, observations and FGDs provided perspectives on key contextual factors and mechanisms shaping CHWs' ability to deliver people-centred care.

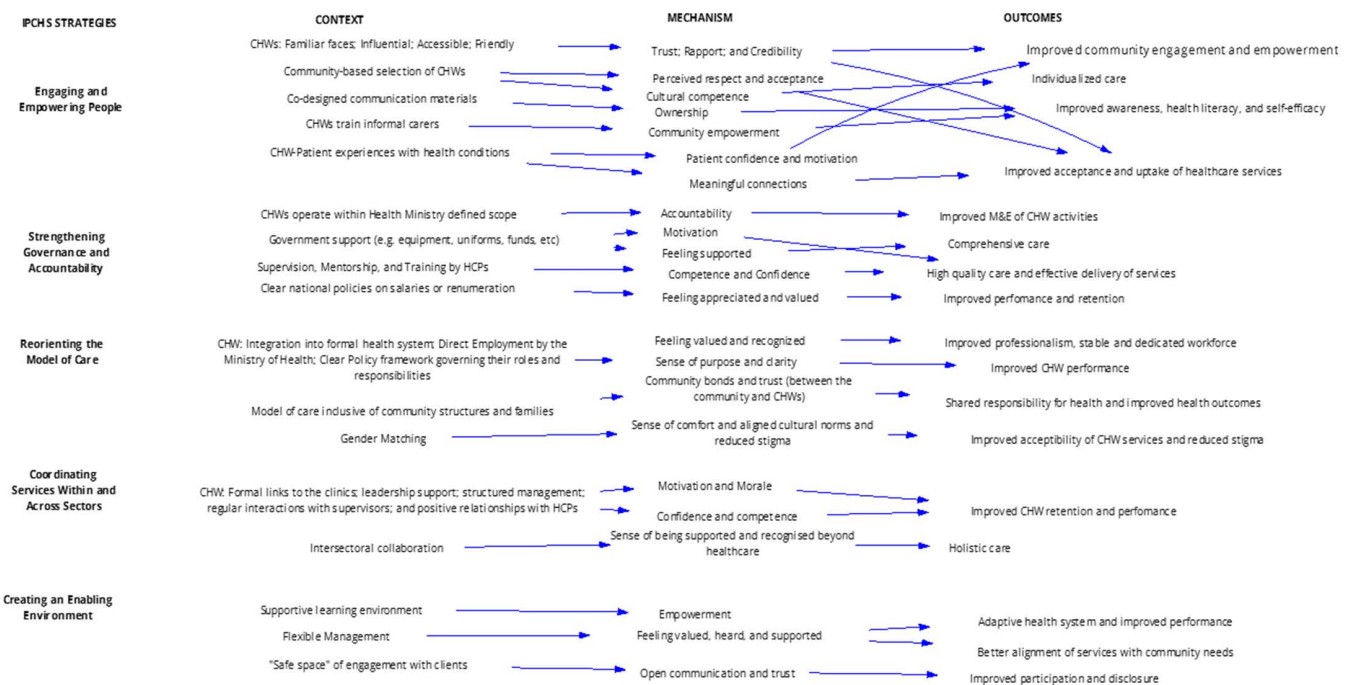

**Fig 2. Schematic representation of the initial programme theory for enhancing CHW roles and responsibilities in service of IPCHS [26].**

Home visits were pre-scheduled by the study team (SB, GR and UB) in collaboration with the CHW, the OTL, and the operational managers (OM) at the respective clinics. SB and GR are qualified nurses, the latter having over 30 years' experience with CHW programmes. Upon confirmation of the scheduled visit, the study team accompanied the CHWs to each household. To avoid altering or interrupting these routine CHW visits, the study team conducted observations and interviews only after the CHW had concluded her household visit.

Structured observations and semi-structured interviews were conducted in isiZulu, audio-recorded, and translated verbatim into English. All transcripts were quality checked following a standardized protocol for data translation and transcription. Initial observations documented environmental characteristics such as the presence of a vegetable garden and other contextual factors before entering the house. In the household, the team observed the interaction between the CHW and the household members assessing if the engagement was people-centred, and identified any other relevant social or contextual dynamics at play using a checklist with an option to add observation notes, guided by the IPCHS framework. Following these observations, individual interviews were conducted with key participants to understand the role, challenges, and opportunities in the work of CHWs to provide people-centred care.

*Focus Group Discussions*: To corroborate the household-level findings obtained from household observations and individual interviews, FGDs were conducted with three of the five WBPHCOTs included in the study. These occurred following routine CHW data reporting sessions to minimize disruption and ensure that CHWs were able to fulfill their professional duties before participating in the discussions. The FGDs focused on CHWs' daily responsibilities and interactions with the communities; community health conditions and social issues; structural and systemic barriers to providing care; collaboration with other health system actors, including NGOs and government departments; the role of CHWs in case management, referrals, and service integration; access to resources, training, and the support CHWs require to perform their duties; data collection, reporting mechanisms; and CHW engagement in health system governance. Each FGD comprised 12–15 CHWs. The FGDs were not limited to CHWs who had been directly observed during home visits. Instead,

**Table 2. Data sources, methods of data collection, and use in analysis.**

| Data Source | Method of Data Collection | Use in Analysis |
|---|---|---|
| CHW-Community Interactions | Structured observations of CHWs during home visits in five rural KwaZulu-Natal communities (n = 15) | Observations assessed household access, CHW-community engagement, and alignment with IPCHS principles. Data informed thematic coding and identified contextual dynamics influencing CHW effectiveness. |
| Service Users (Household Members) | Semi-structured interviews with service users living with two or more chronic conditions (n = 15) | Explored user perceptions of CHWs, health needs, and barriers to care. Data contributed to identifying mechanisms such as trust, confidentiality, and healthcare access. |
| Household Decision Makers (HDMs) | Semi-structured interviews with HDMs identified by the household as decision authorities (n = 15) | Provided insights into household-level social determinants, caregiving roles, and support systems affecting service users. Data was also contextualised to study mechanisms influencing CHW impact. |
| Outreach Team Leaders (OTLs) and Clinic Operational Managers (OMs) | Semi-structured interviews conducted at clinic level (OM) (n = 2) and community (OTL) (n = 2) | Captured supervisory and managerial perspectives on CHW performance, integration into formal health systems, resource gaps, and governance structures. Data triangulated with community-level findings to understand institutional enablers and constraints. |
| Community Health Workers (CHWs) | Focus Group Discussions (FGDs) with CHWs from three different WBPHCOTs (n = 3) and individual interviews (n = 15) | Interviews provided collective reflections on daily responsibilities, barriers to care, inter-sectoral collaboration, resource access, CHW support needs, and contexts that trigger mechanisms for the delivery of people-centred care outcomes. FGDs corroborated household-level findings and expanded insights into systemic and governance factors. |
| Field Observations | Environmental scanning during home visits (e.g., presence of gardens, sanitation conditions) (n = 15) | Contextualised household conditions and community resources to better understand environmental influences on CHW service delivery. |

all CHWs from the clinic were invited to participate. This broader engagement allowed for the triangulation, corroboration, and expansion of the study's household level findings by incorporating insights from a wider range of CHWs. A similar data quality checking process was followed as that of the interviews.

## 2.4. Step 3: Data analysis and refining the programme theory

Data analysis followed a systemic, multi-step approach to understand CHWs' roles and responsibilities in delivering people-centered care within the IPCHS framework. The process began with thematic coding, where qualitative data was systematically categorized to identify recurring themes. This was followed by a case analysis, which enabled an in-depth examination of individual cases to contextualize variations and commonalities across different CHWs, users, and household leader experiences.

This was followed by matrix coding to systematically compare and contrast themes across multiple cases in NVIVO [28], facilitating a deeper exploration of relationships within the data. To enhance analytical rigor, DeepSeek AI (Version DeepSeek-R1) was integrated into the data analysis process, providing computational support for pattern recognition and identification of CMO configurations [29]. Firstly, transcripts were uploaded into an AI tool followed by a prompt to identify existing context-mechanisms and outcomes. This was followed by applying the R1 reason feature. A DeepSeek AI feature designed to simulate human-like reasoning. This was done to identify and articulate plausible connections between contexts, mechanisms, and outcomes that exist within the data, and provide traceable explanations for why certain interpretations were generated, allowing the research team to assess, validate, or refine emerging CMO patterns.

The findings from an AI tool were further refined through expert consultation (via regular meetings with the research team to discuss CMOs), literature synthesis, and field experience, ensuring that emerging insights were aligned with

the existing knowledge and practical realities. This iterative process supported the development of demi-regularities, where we identified recurring but context-dependent patterns in CHW practices and interactions. These insights contributed to the refinement of the programme theory and development of middle-range theories, bridging empirical observations with broader theoretical constructs to explain how CHWs deliver people-centered care in diverse settings. Specific examples of the prompts used, the outputs generated by the R1 Reason feature, and the process by which these outputs were reviewed and validated by the research team using realist logic are provided in supplementary file (S2 File).

In addition, outcomes were defined as the observable or expected effects that followed the activation of mechanisms within a specific context [21]. These included behavioural, institutional, and systemic changes that reflected shifts in practice or system functioning. Outcomes were initially identified through inductive thematic coding [30], retroduction [22,31], and AI-supported pattern recognition, and were then refined and validated in team synthesis meetings where distinctions between context, mechanism, and outcome were tested against realist logic and empirical data.

This rigorous analytical approach ensured a comprehensive understanding of CHWs' roles and functions, enabling the development of actionable insights for strengthening community health programmes following these seven steps: 1. Thematic coding. 2. Case analysis, 3. Matrix coding, 4. AI integration, 5. Expert, literature, and field experience, 6. Theory development, and 7. Programme refinement as illustrated in Fig 3.

## 2.5. Ethics statement

Ethical approval was obtained from the **Biomedical Research Ethics Committee (BREC)** at the University of KwaZulu-Natal **(BREC/00002768/2021)**. Written informed consent was obtained from all participants prior to data collection. The use of audio recordings and observation notes was carefully managed to ensure that no identifying information was disclosed in the transcripts or analysis.

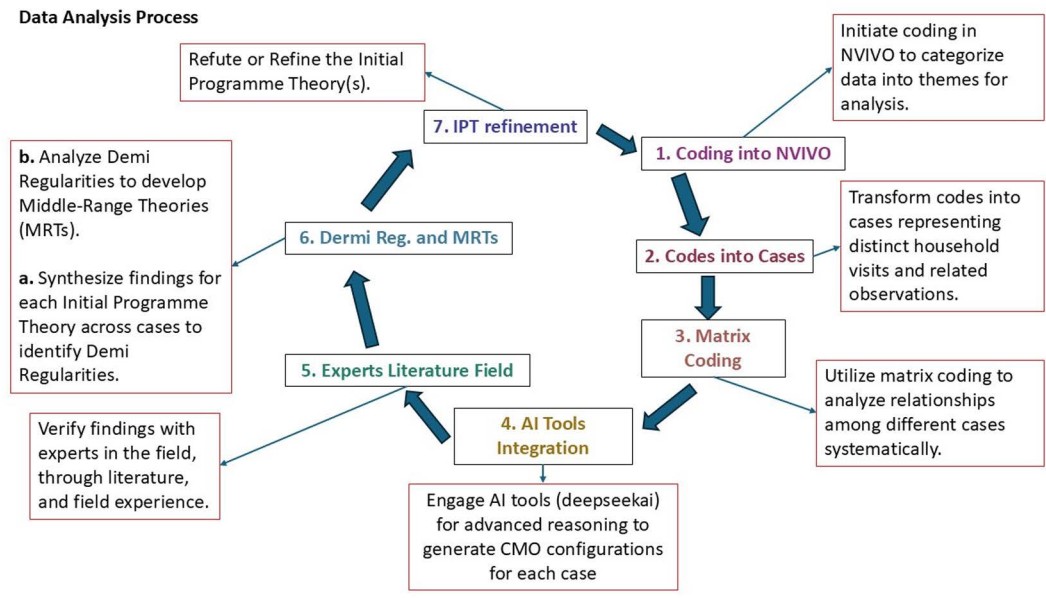

**Fig 3. Data analysis flow process.**

## 3. Results

The study applied the CMO framework to explore how CHWs operate (roles and functions) in different communities to deliver people-centred care. Guided by the IPCHS framework strategies, the following sections outline the interaction between context and mechanisms and how these interactions produce outcomes.

### 3.1. Macro-level CMO configuration for CHW community engagement and empowerment IPCHS strategy and proposed middle-range theory

**3.1.1. Context.** CHWs are critical in delivering people-centred care in resource-limited settings, where individuals often face structural and social barriers such as stigma, financial constraints, and household restrictions. As members of the same communities they serve, CHWs must navigate complex social dynamics to gain trust, ensure confidentiality, access households, advocate for patients, and provide material support. These mechanisms determine whether individuals engage with healthcare services through CHWs or avoid them, shaping health outcomes.

**3.1.2. Mechanisms.** *Trust Formation and Confidentiality*: Trust and confidentiality are central to CHWs' effectiveness, shaping whether individuals disclose health concerns. CHWs who demonstrate warmth, cultural sensitivity, and approachability foster trust, enabling open discussions about sensitive issues. As one participant noted, *"If you are a community health worker you must be approachable so that you can work well with people, and so they can open up to you about their secrets."* (GME-FGD). Another added, *"A community health worker is said to be down to earth; when we say here is this household sit down on a mat, a community health worker must sit down on a mat."* (GME-CHW-Case 1). CHWs' role as intermediaries is valued, with community members relying on them for healthcare access and emergency assistance. For example, *"…They think that the ambulance will arrive quicker if it is called by the community health worker."* (TYS-FGD). However, perceptions of CHWs as volunteers rather than professionals sometimes limit disclosure, while breaches of confidentiality—real or perceived—severely undermine trust, leading to secrecy and avoidance of care. One CHW explained, *"One thinks that you are still a volunteer and never gives full information."* (BUE-CHW-Case 1). Furthermore, fear of judgment and gossip discourages engagement, with some viewing CHW visits as *"Nagging."* (BUE-HDM-Case 1) And *… people don't want to let us into their homes because we gossip."* (GME-FGD). Therefore, balancing persistent follow-up with community receptiveness and confidentiality remains critical to maintaining trust and ensuring healthcare access.

*Familiarity as a Double-Edge Sword in CHW Engagement*: CHWs' familiarity with the communities they serve has a dual impact on healthcare engagement, acting as both a bridge and a barrier. On one hand, shared backgrounds foster trust, making individuals more willing to disclose health concerns, as local CHWs are seen as understanding their lived experiences. As one participant noted, *"If you enter this household with a community health worker from this area, that is the only time you will get responses."* (GME – FGD). On the other hand, this same closeness can create discomfort, with individuals—and even CHWs themselves—hesitating to discuss sensitive issues due to fears of breached confidentiality. One CHW explained, *"One doesn't like that—maybe let's say a CHW from across is perhaps a relative of the person I am working with, maybe he doesn't like to be known."* (CAD-CHW-Case 2). Similarly, another added, *"I wouldn't want to visit my neighbor because it would feel too personal."* (GUA-CHW-Case 2). This tension highlights the need for balanced strategies that leverage trust while safeguarding privacy to maximize CHW effectiveness.

*Household Access as a Gatekeeper to CHW Effectiveness*: Household access is a critical factor in determining the effectiveness of CHWs, shaping both healthcare delivery and community trust. When CHWs are welcomed into homes, they provide not only medical support but also emotional reassurance, reducing feelings of isolation and fostering engagement. One respondent shared, *"Yes, community health workers' home visits are really good… when she shows up, I'm often happy that she's here to help"* (TYS-USER-Case 1), and *"Even though she just comes and sits with us and chats… we are relieved… we are not abandoned"* (TYS-USER-Case 1). However, access is not guaranteed—stigma, privacy concerns, and distrust often lead households to refuse visits, particularly for sensitive conditions like HIV. CHWs

must navigate these barriers with persistence and humility, yet restricted access leaves gaps in care, increasing the risk of untreated illnesses. One CHW noted, "*Maybe one doesn't want to be known at home, like in some families they still believe that if you have HIV, you live a careless life* (CAD-CHW-Case 1), while another added, *It is said that so and so can't enter, let's try to be humble and plead to be able to enter that household*" (GME-CHW-Case 1).

The contrast between households that embrace CHWs and those that resist them underscores the dual challenge of building trust while overcoming deep-seated fears and misconceptions that undermines household access.

*Healthcare Advocacy and Navigation*: CHWs play an important role in bridging gaps in healthcare access, particularly for marginalized individuals facing financial, logistical, and systemic barriers. Trusted CHWs act as advocates, helping patients navigate complex healthcare systems, secure medications, and communicate with providers—directly improving treatment adherence and reducing fear of medical visits. As one patient noted, *"She speaks on our behalf at the clinic."* (GME-USER/HDM-Case). Another explained how CHWs help overcome hesitation: *"Yes, maybe she would say that, well, I have missed [my appointment] and I am scared to fetch it [medication]. I would then say maybe I will fetch it."* (BUE-CHW-Case 1). However, when trust or access to CHWs is lacking, individuals often delay or avoid care, increasing risks of untreated illness and worsening health outcomes. Financial constraints, such as transportation costs, further exacerbate these barriers, though CHWs can provide alternatives—as one user shared: *"...I don't have to wait until I have R50 to go to the clinic. I can just talk to the community health worker and they might have a solution to help me."* (GME-CHW-Case 2). This underscores the need for stronger institutional support to sustain CHWs' critical role in ensuring equitable healthcare access.

*Material Support as a Reinforcer of Trust and Household Access*: The study reveals that material support is a critical factor in shaping CHWs' credibility and fostering trust, as it demonstrates their tangible commitment to community well-being. When CHWs provide essential supplies—such as food, diapers, and medication—community members view them as both healthcare advisors and practical allies, increasing engagement and caregiver confidence. As one participant noted, *"Well, that's what the community health workers are doing…to look out for us, bring us pampers (adult diapers) and gloves… and also check how we are taking care of the patients. She comes and looks."* (TYS-HDM-Case 3). However, unmet expectations due to resource shortages lead to frustration, eroding trust and discouraging participation. Discrepancies between anticipated and received aid, as seen in cases of inadequate food parcels, provoke agitation and reluctance to cooperate. One CHW explained, *"They had seen big bundles of food parcels on TV… But here we are given enough for food parcels that were enough for five people [too few for the need of the community] … In some cases, you try and help, and not realise that you are putting yourselves into a trap… I told my colleagues that we should leave because a lot of people suddenly came and they were agitated."* (TYS-CHW-Case 3) Another added, *"People always yearn for us to visit them with parcels in hand. Sometimes we don't have them."* (GME-CHW-Case 3) A CHW echoed this sentiment highlighting material expectations from the community, saying, *"He will ask you – will I get? … because those who came gave me 1, 2, and 3."* (CAD-CHW-Case 1).

These findings highlight the tension between community expectations and the intended role of CHWs. While material support can reinforce trust and credibility, expecting CHWs to routinely provide such resources risks overburdening them. Rather than positioning CHWs as direct providers of material aid, these expectations should be addressed through clearer role definitions, improved referral systems to social services, and more responsive institutions to reduce the need for CHWs to fill these gaps and become de facto providers of material support. [Fig 4](link) illustrates the refined programme theory for community engagement, emphasizing how trust, confidentiality, household access, and material support shape CHWs' ability to deliver people-centred care.

### 3.2. Middle-range theory for reinforcing CHWs' engaging and empowering communities' strategy in IPCHS framework

The evidence presented under engaging and empowering people and community's strategy (section 3.1) supports the following middle-range theory:

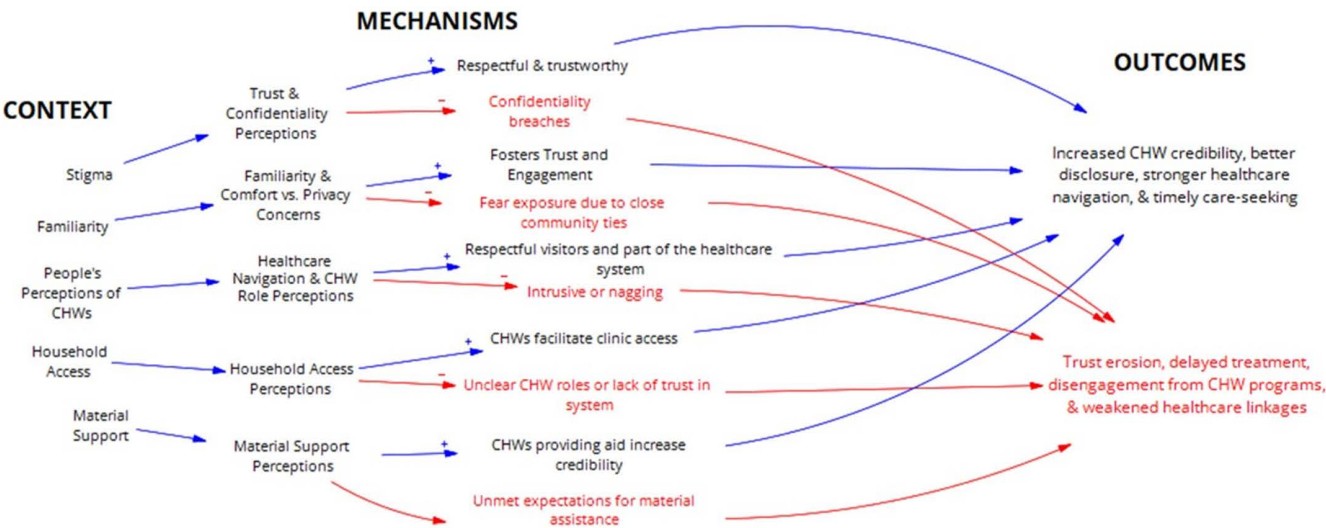

**Fig 4. Refined Programme Theories for Community Engagement IPCHS strategy to deliver Integrated and People-Centred Care through CHWs.**

The effectiveness of CHWs in delivering people-centred healthcare services (engaging and empowering people and communities' strategy) is determined by their ability to establish trust, maintain confidentiality, secure household access, advocate for patients, and provide material support. When these mechanisms align, CHWs can enhance healthcare engagement, disclosure of sensitive conditions, and early treatment access. However, when one or more of these elements' breakdown, CHWs effectiveness diminishes, leading to delayed care, reduced corporation and poor health outcomes. This theory underscores the interdependence of social, structural, and relational factors in shaping CHWs' impact. Addressing contextual challenges and reinforcing key mechanisms can optimize CHW-led interventions, ensuring greater engagement and empowerment of CHWs and the community to improve healthcare access for the community.

### 3.3. Macro CMO configuration: Strengthening governance and accountability for CHWs in delivering IPHCS

**3.3.1. Context.** CHWs face precarious employment conditions, lack professional recognition, and have inconsistent access to essential resources. Their temporary contracts contribute to financial insecurity, and inconsistent uniform provision, and lack of standardized supplies undermines their legitimacy. Additionally, logistical barriers, such as inadequate transport and limited feedback on data collection, further hinder their effectiveness and motivation.

**3.3.1. Mechanisms.** *CHW Integration into the Formal Health System – Job Security and Permanent Contract*: CHWs frequently face precarious working conditions, including temporary contracts, inadequate resources, and a lack of professional recognition, which undermines their motivation and long-term integration into healthcare systems. Many CHWs express frustration over being perpetually stuck in contractual roles without clear pathways to permanent employment, as participants noted: *"We have been working for years, but they still tell us that we are on contracts. When will we be recognized?"* (GUA-FGD) And *"…they are contract workers... I feel they must be permanent."* (CAD-OM-Case 3). Others highlight the absence of essential materials, training, and even uniforms—symbols of legitimacy and respect—with one stating, *"We need materials to work. We need more in-service trainings... We need better cars. We also need uniform, so that CHWs can be properly recognisable."* (GUA-OTL-Case 3). This instability fosters a sense of being undervalued, with some describing themselves as *"stepping stones"* for the health system rather than integral contributors. A CHW stated, *"Because we are also workers under the department they must register us and don't be hypocrites, and*

*make us step ladders for the Department of Health please.”* (GUA-FGD). Without addressing these systemic gaps, efforts to formalize and sustain CHW programmes risk falling short, limiting their potential impact.

**CHW Recognition by Healthcare Professionals, Community, and Stakeholders -Professional Legitimacy and Recognition**: CHWs lack formal recognition and visible markers of authority, leading to ongoing misconceptions about their role within the healthcare system. Without uniforms or professional titles, they are often mistaken for volunteers, reducing their ability to gain trust from community members and healthcare staff. CHWs strongly believe that wearing uniforms would enhance their legitimacy, with one stating, *“And the uniform will be a symbol that you are permanently employed for this job.” (GUA-CHW-Case 3)* Another CHW explained how a lack of professional attire diminishes their standing in the community: *“If we had a uniform, people would see that we are health workers and not just community members… “Yes, maybe uniforms will give us dignity.” (BUE-CHW-Case 1)*. The absence of official identification and professional markers continues to contribute to negative community perceptions of CHWs and limits their influence in public health interventions.

**Resource Shortages & Perceived Neglect**: A critical governance challenge is the inconsistent provision of essential resources for CHWs, creating disparities in service quality and undermining their credibility. While some CHWs reported receiving basic supplies like gloves, masks, or uniforms, others in similar catchment areas lacked these fundamental tools, forcing them to purchase their own. One CHW noted, *“Yes, we receive gloves and masks… We received jerseys and t-shirts/tops”* (BUE-CHW-Case 3), while another explained, *“When it comes to working supplies, they are very short. We have to buy the exercise books that we use for ourselves. They are not provided to us. You have to find a way to replenish anything that runs out. As I have mentioned, materials such as gloves, all the materials that I need to assist in the households where there are challenges”* (GME-CHW-Case 3). This inequity reinforced feelings of neglect and eroded community trust, as CHWs without proper equipment were perceived as less professional. As one CHW stated, *“If we wear our own clothes, we are taken for granted”* (BUE-CHW-Case 1). The absence of a standardized supply chain system not only hindered service delivery but also reflected systemic undervaluation of CHWs, leaving them unable to perform their roles effectively.

**Fair Compensation and Benefits for CHWs**: Financial insecurity undermines the stability and effectiveness of CHWs, who face heavy workloads and emotional demands while receiving inadequate compensation. Many CHWs enter the field driven by a sense of purpose, but persistent low wages—often without benefits—lead to frustration, attrition, and a perceived loss of dignity. As one CHW stated, *“We are not getting paid sufficiently… Another thing that I have noticed about CHWs is that they are stripped of their dignity. For example, you may find that here in X Clinic, just because I am a CHW, and she has red epilettes… I am nothing. I am nothing when she starts to speak”* (GUA-CHW-Case 2). Wage disparities between CHWs and formal healthcare workers further deepen resentment, fueling turnover as CHWs seek better-paying alternatives. Another CHW expressed this frustration, saying: *“I will look for another job because the money is little here—maybe it's little and it doesn't satisfy me.”* (BUE-CHW-Case 1). This disrupts care continuity and weakens community health programmes, highlighting the urgent need for fair wages, benefits, and job security to sustain CHW retention and service delivery.

**Transparency and Feedback Mechanisms in Data Collection**: Another critical governance failure relates to the lack of transparency in data collection and feedback mechanisms. CHWs are responsible for gathering extensive patient and household health data, yet they rarely receive updates on how this information is used in decision-making or program design. Many CHWs reported feeling that their efforts were ignored, reducing their enthusiasm for data collection. One participant described this frustration, stating, *“We submit that report, and the Supervisors take it to the Manager… we don't know what happens then”* (CAD-CHW-Case 2). Community members also expressed distrust toward repeated surveys, feeling that data collection efforts did not translate into tangible healthcare improvements. This weakened cooperation, as some individuals became reluctant to share information. One community member voiced this concern, stating, *“So this sister said that there was someone who worked here and asked us these things you are now asking us… don't you*

*take this information and put it in the system?" (BUE-CHW- CASE 1).* Establishing real-time digital data-sharing systems where CHWs receive feedback on how their data informs health policy could enhance CHW engagement with the health system, foster motivation, and strengthen trust between CHWs and communities.

***Logistical Support and Infrastructure for CHW Programmes***: Transport barriers significantly hinder CHWs, limiting their reach and effectiveness. Many CHWs must walk long distances or use personal funds due to unreliable government transport, delaying home visits and follow-ups. This not only strains their ability to report cases promptly but also leads communities to perceive delays as neglect, damaging trust. For instance, CHWs highlighted the need for transport support to reach distant patients and training sessions. One noted, *"…if we could say there's a transport of some sort that takes us to far apart places because one sometimes walks for a long distance to work far away"* (BUE-CHW-Case 3). Another added, *"Perhaps what I would like to ask is that maybe if there are Workshops [hmm] they should meet us half way that maybe they provide us with transport"* (CAD-CHW-Case 2). A third CHW emphasized the difficulty of accessing critical reporting hubs such as the war room (An intersectoral platform where NGOs, Private sector, Government institutions, and Communities converge to identify problems and offer solutions that address the community's needs), stating, *"There is no transport this side. I wish they could consider that CHWs must be provided with transport when they are going to the war room, because the war room is where all cases are reported"* (TYS-CHW-Case 3). Without systemic improvements, these mobility constraints will continue to undermine CHW efficiency and community health outcomes. As shown in Fig 5, the programme theory on governance and accountability highlights how employment conditions, professional legitimacy, and feedback systems influence CHW motivation and system performance.

### 3.4. Middle-range theory: Governance & accountability for people-centered CHW programs

**Proposition:**

When governance structures provide job security, professional recognition, equitable resource allocation, transparent data utilization, and logistical support, CHWs experience a greater sense of value, commitment, and legitimacy in their roles. This, in turn, triggers mechanisms such as improved morale, enhanced trust in the healthcare system, and greater accountability to both communities and policymakers. This subsequently results in a more stable and motivated CHW workforce, increased healthcare service efficiency, and stronger community engagement. However, when governance and accountability mechanisms are weak or inconsistent, CHWs face demotivation, attrition, and reduced healthcare effectiveness, leading to gaps in community healthcare delivery.

### 3.5. Macro CMO configurations for reorienting the model of care in CHW programmes to deliver IPCHS

**3.5.1. Context.** The IPCHS reorienting the model of care strategy advocates for transitioning from an inpatient, facility-based approach towards a community-based model of care that focuses on health promotion and prevention. However, the expanded role of CHWs within this model is constrained by structural and systemic barriers that compromise their performance, credibility, and long-term sustainability. As healthcare systems increasingly prioritize prevention, health promotion, and home-based care, CHWs are tasked with delivering essential medical services alongside social support, such as grant applications, birth registration, and housing assistance. Yet, inadequate structural support, unclear role definitions, and unsustainable workloads hinder their ability to provide consistent, high-quality care. Additionally, weak leadership engagement diminishes their credibility, fueling skepticism among communities about government health initiatives.

**3.5.2. Mechanisms.** ***CHWs' Expanding Roles and the Need for Structural Support***: The expansion of CHWs' roles into social services—such as assisting with grant applications, IDs, and housing—has created a tension between their healthcare duties and community expectations. While CHWs recognize the importance of these social supports, their increasing involvement has led to role conflict, excessive workloads, and burnout. As one CHW explained, *"We are involved with the municipality, Home Affairs, children who don't have certificates, and people who don't have IDs, SASSA."* (BUE-CHW-Case 3).

Communities, facing inaccessible government services, heavily rely on CHWs to fill these gaps, diverting attention from core healthcare tasks. Many CHWs report feeling overwhelmed, with some managing up to 80 households, an unsustainable workload that compromises care quality. *"80 households is undoable,"* (BUE-CHW-Case 3) one noted. Frustration is compounded when urgent social cases, like assisting a raped child, pull them away from medical responsibilities, leaving gaps in healthcare delivery. *"They will say that the community health worker is absent, she's attending to her case, and maybe it's a case of a raped child,"* (BUE-CHW-Case 1) another shared. These challenges underscore the need for clearer role boundaries, workload adjustments, and stronger referral systems to ensure CHWs can prioritize health services without neglecting critical social needs.

***Community-Based and Home-Based Care***: Reorienting the model of care from curative, facility-based services towards health promotion and preventive care requires CHWs to deliver home-based services that support early intervention, continuity of care, and improved self-management. While tasks such as medication collection, delivery, and follow-up may resemble curative outpatient services, in this context they serve a health-promotive function by improving treatment adherence, preventing disease progression, and reducing avoidable hospitalizations. These expanded responsibilities, alongside health education and screening, reflect a shift towards community-based care, but also raise concerns about role strain, burnout, and long-term sustainability.. The other concern is that CHWs are expected to compensate for gaps in the healthcare system and social services, without adequate support, leading to their ability to maintain service quality being compromised. Many CHWs reported experiencing excessive workload pressures, with one stating, *"To deliver it to them maybe we are really going to fail [laugh], now this is going to defeat us." (BUE-CHW-Case 1).* While participant perceive collecting medication as the responsibility of CHWs. One community member said: *"If you don't have tablets you are able to send your community health worker to bring some for you if you aren't supposed to see the doctor…" (BUE-USER-Case 2).* Some CHWs have attempted to set boundaries, prioritizing support for elderly patients while encouraging younger individuals to take responsibility for their own healthcare needs. One CHW noted, *"I fetch for the grannies and I tell the youth to collect it themselves." (BUE-CHW-Case 1).* The lack of clear guidelines on workload distribution

**Fig 5. Refined programme theories for Strengthening governance and accountability IPCHS strategy to deliver integrated and people-centred care through CHWs.**

exacerbates these challenges, increasing the risk of exhaustion and attrition among CHWs. Furthermore, the lack of clarity on the roles and functions of CHWs by the community leads to unmet expectations. When asked about the roles and functions of CHWs, community members had different responses that indicated a lack of Clarity: *No, we see them at the… at the clinic…* (GUA-USER/HDM-Case 3); "The CHW, besides advising us, as I have mentioned, you can quickly contact her if someone falls ill in the household". (CAD-USER-Case 2); and "Social worker? No, this is the one." (TYS-USER/ HDM- Case 2). Therefore, establishing clear guidelines and effectively communicating them to both CHWs and the community is essential for sustaining healthcare support and preventing CHW work overload. This ensures people-centered care by aligning expectations between both parties.

*Leadership Gaps and Community Trust*: The absence of visible leadership and direct engagement from senior health officials has eroded community trust in healthcare services, leaving CHWs to face skepticism alone. CHWs report that community members often question the legitimacy of government health programmes, asking why higher-level officials never visit them. When you go to that household, they will say that we have never seen your seniors whom you always mention (BUE-CHW-Case 1). Similarly, CHWs explain, "*We tell them that we work under the government, but they ask, 'Where are the people from the government? Why do they not come here?'*" (BUE-CHW-Case 1). This disconnect undermines CHWs' credibility and community engagement. However, when the OTLs are actively involved—providing guidance, resources, and clinic referrals—CHWs feel more supported and communities gain confidence in their work. One CHW noted, "*Our OTL was helping because he was—he would ensure that we have all the resources to check TB, those who don't like going to the clinic would be checked at home…I would also talk to the OTL if I found someone whom I see is not in good condition…the OTL would go and maybe we would go with him there*" (CAD-CHW-Case 1). Another observation was made where "an *OTL observed the patient's diabetic wound, providing care and referral to come and see her at the clinic where there are more resources and doctors*" (Observation-GME-Case 3), highlighting how OTLs support CHWs when they are present during home visits. This contrast highlights the critical role of visible leadership. Therefore, to strengthen healthcare delivery, systemic reforms must prioritize consistent leadership presence, formal collaboration frameworks, and institutional backing for CHWs. Fig 6 presents the refined programme theory for reorienting the model of care, detailing how role expansion, workload, and leadership visibility affect CHWs' capacity to deliver preventive and home-based services.

## 3.6. Middle-range theory for reorienting the model of care

**Proposition:**

When healthcare shifts from an inpatient, facility-based model to a community-based model of care, the role of CHWs expands beyond traditional healthcare delivery to include social support services. This expansion is driven by community trust, accessibility needs, and systemic gaps in healthcare infrastructure. However, the absence of clear guidelines, leadership engagement, and workload management strategies leads to role conflict, dependency formation, and eventual burnout. CHWs, in response, develop adaptive strategies such as informal boundary-setting, advocacy for policy reforms, and efforts to renegotiate their roles within the healthcare system.

## 3.7. Macro CMO configuration for coordinating services within and across sectors

**3.7.1. Context.** The findings highlight critical gaps in cross-sectoral coordination, affecting both community members' access to essential services and CHWs' ability to function effectively. Across government agencies, NGOs, and health services, poor institutional responsiveness, fragmented service coordination, and bureaucratic barriers limit the effectiveness of community-based healthcare interventions.

CHWs frequently refer cases to government institutions such as SASSA, Home Affairs, and the police, expecting timely intervention. However, these referrals often go unanswered or experience significant delays, leaving CHWs with no formal mechanisms for follow-up. Additionally, NGOs operate independently from government services, leading to duplication of efforts in some areas while other communities remain underserved.

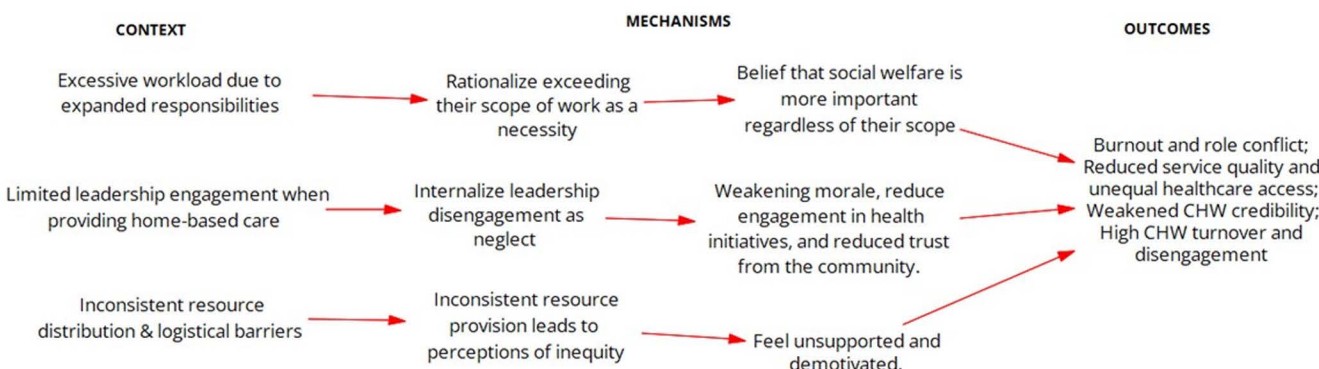

**Fig 6. Refined programme theories for Reorienting the Model of Care IPCHS strategy to deliver integrated and people-centred care through CHWs.**

For vulnerable populations, bureaucratic hurdles, particularly the lack of identification documents, create systemic exclusion from healthcare, education, and social grants. CHWs often fill the gap by assisting individuals with documentation applications, yet they lack formal training or institutional support, adding to their workload.

The interplay of institutional failures, lack of coordination, and bureaucratic inefficiencies has led to a growing reliance on CHWs, traditional leaders, and NGOs as alternative service providers, weakening public trust in formal institutions.

**3.7.2. Mechanisms.** *Frustration and Erosion of Trust in Government Services*: Repeated government non-responsiveness has eroded trust in formal institutions, pushing communities to rely on informal networks and overburdening CHWs. Systemic failures—such as unanswered reports, absent stakeholders in coordination platforms, and bureaucratic obstacles—leave CHWs to fill gaps in social services, from securing IDs to advocating for grants. This unsustainable expansion of their roles fuels frustration and burnout, as their efforts are often met with institutional inertia. As one CHW noted, *"Most problems that we face is that we report most issues and never receive replies"* (GME-CHW-Case 1). Another added, *"We fight (verbally) with SASSA; we bring our applications, people don't have IDs, people don't get their children's certificates because of the network that you say is not available. We are desperate; there's no electricity"* (GME-CHW-Case 1). When systemic neglect persists, even designed solutions like "War Rooms" become ineffective, deepening disillusionment among both CHWs and the communities they serve. *"You see, yesterday we were in the war room. When you table a problem, what makes it difficult is that most departments do not attend"* (GUA-CHW-Case 2). The burden on CHWs grows ever heavier, as one participant explained: *"A community health worker is involved in all sectors; there's nowhere she's left out"* (BUE-CHW-Case 3). Over time, the lack of meaningful change leads to disengagement, with one CHW stating, *"You keep attending this war room until you get tired and decide to stop attending because people are not getting helped anyway"* (TYS-FGD).

*Service Fragmentation and CHW Disempowerment*: The lack of structured collaboration between NGOs and government health services has resulted in fragmented healthcare delivery, undermining CHWs' effectiveness and eroding community trust. CHWs report that NGOs often bypass local structures, implementing programmes without consultation, which leads to overlapping services in some areas and gaps in others. As one CHW noted, *"Sometimes they are disturbing, especially if the NGO will not work together with us. It's better if the NGO will arrive and work with us so that whenever there is a problem, the community will blame you…"* (CAD-CHW-Case 1). Another participant added, *"Other stakeholders or NGOs use us for their goals to be successful"* (GUA-FGD). This disjointed approach not only diminishes CHWs' credibility but also disrupts care continuity, as patients treated by NGOs are frequently misclassified as defaulters in government systems. A CHW explained, *"When people start taking their treatment from them, they default at the clinic.*

*That is the second problem that we have... We are not even reaching our national targets because of these NGOs that the government allows to come in and work with clinics"* (TYS-FGD). Furthermore, uncoordinated NGO interventions sometimes displace government-trained CHWs, demoralizing them and weakening community health programs. One CHW shared, *"We went and got trained... but while we were waiting for that to happen, we saw an NGO coming in to do the exact same thing that we had been trained to do, and we were shut down just like that"* (TYS-FGD). Without shared data systems or joint planning, these inefficiencies persist, leaving both CHWs and communities caught in a cycle of instability and mistrust.

***Bureaucratic Barriers Reinforce Social Exclusion***: The findings highlight how bureaucratic inefficiencies, and institutional barriers disproportionately exclude undocumented immigrants and their children from accessing essential services, perpetuating cycles of poverty and marginalization. Without identity documents, individuals cannot secure social grants, education, healthcare, or formal employment, leaving them dependent on overburdened CHWs and informal networks. As one CHW explains, *"Some of these children's parents come from Lesotho, you see. You tell the person that they must go and apply for a birth certificate for their child, only to find that even the parent doesn't have an ID."* (GME-CHW-Case 2). Home births further complicate documentation, as children delivered outside hospitals face prolonged administrative hurdles. Another CHW notes, *"…it happens that she delivered here at home and went for the BCG here (local clinic), that child cannot be issued a certificate as a child that was delivered at the hospital. Yours as a community health worker is to refer them to different sectors… a community health worker's job will be to refer that child to SASSA."* (GME-CHW-Case 1).

CHWs, though pivotal in bridging gaps, lack formal training and government support to navigate these systemic inefficiencies, leading to frustration and eroded trust in institutions. One unattributed CHW statement captures this struggle: *"People come to us for help, but we also struggle with the system."* Another adds, *"Sometimes you report the same issue repeatedly, and you get told that they are still conducting research and following up on the issue that you reported [laughs]."* (GME-CHW-Case 2). The reliance on CHWs and traditional leaders underscores the failure of formal systems to respond to marginalized communities, placing unsustainable demands on local actors who lack the authority or resources to enact meaningful change. Fig 7 displays the programme theory for service coordination, revealing how systemic fragmentation and limited intersectoral collaboration impact CHWs' effectiveness in navigating and linking communities to social services.

### 3.8. Proposed middle-range theory for strengthening cross-sectoral coordination to improve CHW effectiveness

**Proposition:**

The effectiveness of CHWs in delivering integrated, people-centered care hinges on institutional responsiveness, structured service coordination, the removal of bureaucratic barriers, and CHW empowerment through policy integration and institutional support. When government agencies, NGOs, and CHWs collaborate effectively, service delivery becomes timely, efficient, and accessible, reducing community reliance on informal networks. Additionally, recognizing and fully integrating CHWs within formal health and social systems enhances their ability to coordinate care, advocate for patients, and navigate administrative barriers, ensuring more streamlined support for communities. However, when institutions fail to respond, services become fragmented, and bureaucratic challenges continue to prevent vulnerable populations from accessing essential care, and CHWs are left to bridge these gaps, leading to them becoming overburdened. As a result, community trust in formal systems declines, and health inequities persist.

### 3.9. Macro CMO configurations for creating an enabling environment in CHW programmes to deliver IPCHS

#### 3.9.1. Responsiveness and flexibility of CHWs. *Context*: CHWs are allocated a specific number of households but often extend their responsibilities beyond their assigned duties. They navigate logistical challenges such as transport constraints and time limitations while also serving as key intermediaries between communities and formal healthcare

services. Limited formal training further necessitates peer-to-peer learning, while stigma and fear in clinical settings create barriers to open patient communication. Home visits, therefore, play a crucial role in fostering safe spaces for patient engagement.

*Mechanisms*: CHWs demonstrate a profound commitment to equitable healthcare, often exceeding their formal household allocations to address unmet needs, despite the personal and financial burdens this entails. As one CHW explains, *"We are allocated 60 households. However, we end up reaching 100 households because we want everyone to get this information."* (GME-CHW-Case 3). Their work is also marked by personal sacrifice, with another noting, *"Things like soap, you can get from your own house, but this ends up hurting us... but it is because you want their situation to improve."* (GME-CHW-Case 3). Their work is characterized by fluid prioritization, as they dynamically respond to urgent cases—ranging from emergencies to social support—even when it disrupts planned schedules. One CHW describes this adaptability: *"They'll ask and say that there's a problem there and you don't refuse—you just leave there and go to assist and see what their need is, then try to make connections with the relevant departments for referral."* (GME-FGD). CHWs see themselves as vital connectors between communities and formal health systems, stepping in as first responders and facilitators of care. While this adaptability strengthens healthcare access, it also leads to emotional strain and burnout, underscoring the need for systemic support to sustain their efforts.

**3.9.2. Supportive learning environment and safe spaces.** *Context*: The lack of formal training opportunities means that CHWs rely heavily on peer-to-peer knowledge transfer. This informal system creates variations in understanding and application of knowledge, as some CHWs become well-informed while others experience gaps. Fear and stigma in clinical settings further prevent many patients from openly discussing sensitive health issues, leading to delayed treatment. Home visits provide an alternative space for patient engagement, fostering trust and openness.

*Mechanisms*: CHWs rely on peer-to-peer knowledge transfer, which fosters hands-on learning but also creates inconsistencies in skills development, highlighting the need for more structured training reinforcement. As one CHW noted, *"Yes, sometimes it happens that one person will be taken, and you—the one who has been taken—you have to come back and teach the others who didn't go."* (CAD-CHW-Case 1) Others described collaborative learning: *"We find ways, maybe during the monthly report, where we remind each other everything that we learned. If I managed to grasp the information easily, I can share it with my colleague where she is not clear."* (GME-CHW-Case 3).

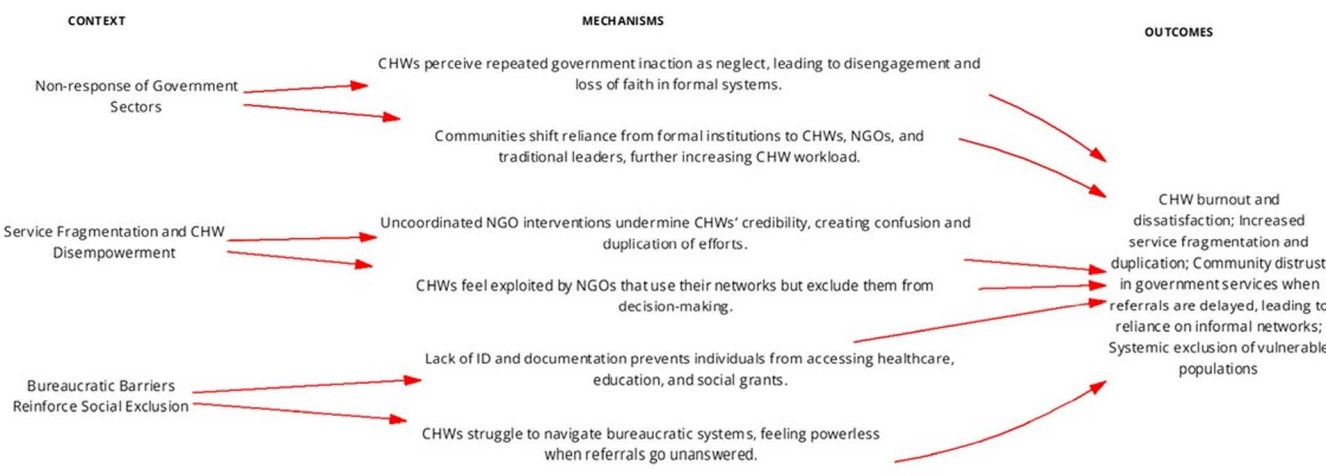

**Fig 7. Refined programme theories for Coordination of Services Within and Across Sectors IPCHS strategy to deliver integrated and people-centred care through CHWs.**

On the other end, patients often avoid clinical settings due to fear of being scolded, leading to delayed care. One participant expressed this fear stating, *"When she comes back she is afraid of going to the clinic to collect because she missed a date, she is afraid because they say they are scolded at the clinic."* (CAD-CHW-Case 1). In contrast, home visits provide a safe space where patients feel more comfortable sharing sensitive issues, reinforcing CHWs' belief in the value of home-based care—though this also deepens their emotional burden. An OTL explained, *"…if the OTL is present together with the CHW, other people open up more when you visit their home to address their problems that have to do with chronic ailments…They open up more than when they would at the clinic…They are scared to open up at the clinic… Because you are now in their natural setting."* (GME-OTL-Case 3).

These insights underscore the importance of leveraging home visits to bridge gaps in trust and disclosure that clinical environments struggle to address. Fig 8 illustrates the refined programme theory for creating an enabling environment, showing how peer learning, moral commitment, and adaptive practice support CHWs in the face of systemic barriers.

### 3.10. Middle-range theory for creating an enabling environment for CHWs to deliver integrated and people-centred care

**Proposition:**

The ability of CHWs to provide integrated, people-centered healthcare is shaped by their moral commitment, adaptability, and peer-supported learning but is constrained by systemic structural barriers, workload pressures, and role ambiguity. When CHWs perceive their role as a moral duty rather than a defined job, they extend their services beyond formal allocations, leading to increased healthcare access but also greater emotional and physical strain. Similarly, in environments with limited formal training and stigma-related healthcare avoidance, CHWs rely on peer-to-peer knowledge transfer and home visits to create safe spaces for learning and patient engagement, improving trust but also leading to inconsistencies in knowledge application and increased emotional burden. If structural support mechanisms, role clarity, and training standardization are not implemented, CHWs will continue to navigate these tensions reactively, resulting in burnout, unequal service delivery, and compromised long-term sustainability of community-based care. Therefore, the effectiveness of CHW programmes in bridging healthcare gaps and fostering trust-based care models depends on balancing their intrinsic

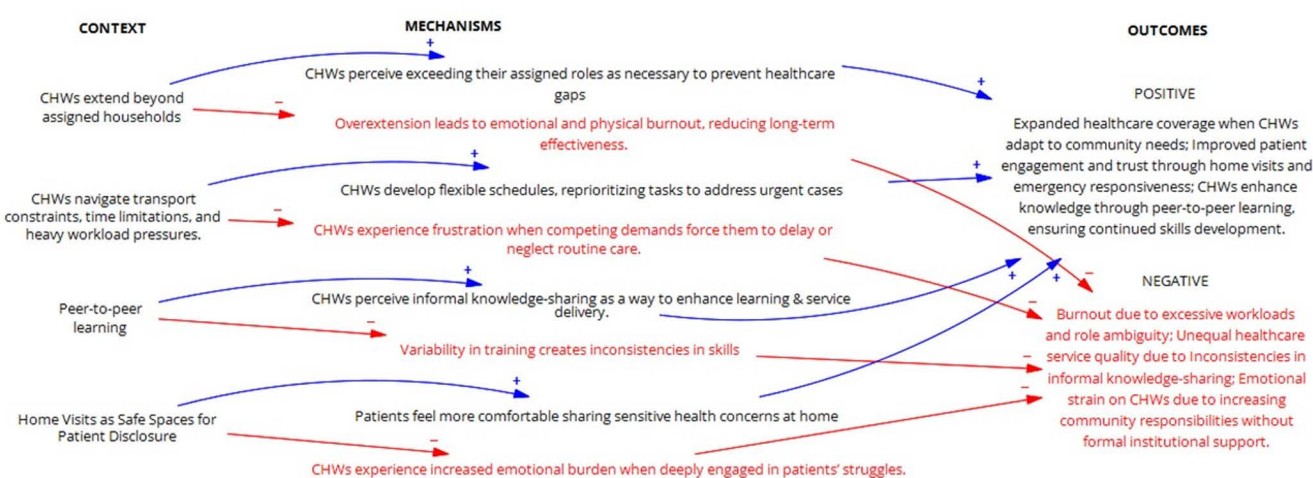

**Fig 8. Refined programme theories for Creating an Enabling Environment IPCHS strategy to deliver integrated and people-centred care through CHWs.**

motivation and adaptability with external structural support, intersectoral collaboration, workload regulation, and formal recognition within the healthcare system.

### 3.11. Refined programme theory

The refined programme theory (RPT) incorporates new findings and adds the necessary level of detail to aspects of the IPT that were underdeveloped. Key refinements relate to governance and accountability, the employment and recognition of CHWs, reorienting the model of care, intersectoral coordination, mechanisms for community engagement, logistical support, and exclusions from the previous framework. One of the key refinements is a broader emphasis on governance and accountability. The earlier programme theory lacked details on data transparency and feedback. The RPT specifically introduces structured feedback systems as a strategy to increase CHWs' motivation and performance.

The Revised RPT on reorienting the Model of Care included detailed examples of how CHWs are now handling social service tasks like helping with grant applications and birth certificate registrations in addition to their healthcare responsibilities. It further emphasized the challenges of transforming the care model from curative and inpatient facility-based care to preventative and home-based care. It highlighted the risks of burnout and unsustainable workloads, which endanger CHW retention and service quality. The RPT also pointed out that the lack of visible leadership from senior health officials weakens CHWs' credibility and hampers their effective functioning within health systems. Initial program theories, like gender-matching, were omitted due to insufficient evidence, leading to a shift in focus towards more supported factors that influence reorienting the model of care. Moreover, integrating CHWs into the healthcare system was reconsidered as a governance matter, and not as a reorienting the model of care strategy, therefore was moved to governance and accountability strategy. The integrated care model and community support structures were moved to the intersectoral collaboration strategy, emphasizing the requirement for coordinated, cross-sectoral actions to boost CHW capacity in improving community health systems. The RPT also omitted broader statements regarding care model reorientation mentioned in the IPT, opting to concentrate on specific contexts and mechanisms identified in the field.

Another key component of the RPT includes employment and recognition of CHWs as professionals. The revised theory explains the importance of job security and professional status in motivating CHWs while pointing out the negative effects of contractual employment leading to burnout and turnover. Mechanisms such as official uniforms and identification have been introduced, solidifying professional legitimacy and trust within communities. The refinements also include better cross-sector coordination.

The RPT also highlights the need for improved cross-sector coordination and underscores the risk of service fragmentation that may occur when CHWs are neglected in NGO-led and government programme initiatives. In addition, there remain bureaucratic hurdles, especially when it comes to ensuring that those who don't have a legal status in the country receive care — which is recognised as one of the main structural barriers preventing access to healthcare and use of social services.

Community engagement mechanisms have also been expanded. Although the IPT addressed trust-building in general, the RPT specifically describes CHW familiarity, cultural sensitivity, and confidentiality as important mechanisms that can enhance patient disclosure and treatment adherence. Furthermore, it recognizes the two-sided nature of familiarity, where social closeness can either enable or hinder access to healthcare due to privacy concerns.

The enabling environment strategy has been integrated to address logistical and resource-based challenges. Transport and medical supplies are now explicitly recognized as factors that influence CHW efficiency and coverage. In addition, the provision of material support (including food parcels, hygiene products) has been highlighted as a mechanism that can act as a reinforcer of CHW credibility whilst unmet expectations can undermine community trust.

Some assumptions derived in the earlier programme theory have been ruled out. The concept that CHWs must always be recruited from within the specific context and through participatory processes has been removed; the RPT acknowledges systemic governance and the formal integration of roles, rather than the recruitment processes. Similarly, the role of

CHWs in training informal carers has been removed, shifting the focus toward improving CHW performance and the delivery of people-centred care rather than expanding their responsibilities. These improvements ultimately yield a better set of conditions that positively or negatively influence the performance of CHWs in delivering people-centred care. By addressing systemic challenges, while reinforcing trust-building mechanisms, the RPT offers a stronger foundation for optimizing the critical role of CHWs in people-centred healthcare systems.

## 4. Discussion

The research adopted a realist evaluation methodology which followed the WHO's Integrated People-Centred Health Services framework to evaluate CHW optimization for delivering integrated people-centred care in resource-constrained, rural South African settings. The research findings present a detailed understanding of CHW performance through the examination of context, mechanisms and outcomes configurations. The programme theories developed through this research reveal the conditions which support or limit CHW success in delivering people-centred services. The following sections present a discussion of the five IPCHS strategies through which key mechanisms and contextual factors influence CHW performance and how these align with, challenge, or extend existing literature.

### 4.1. Engaging and empowering people and communities

This study reaffirms and contributes to the foundational role of trust, confidentiality, and cultural embeddedness as core enablers of CHW-community relationships. Similarly to existing literature [32,33], our findings show that CHWs' social familiarity and lived experience foster trust, which in turn facilitates disclosure and service uptake. Yet, echoing concerns raised by Grant et al., 2017 and Watkins et al., 2021 [34,35], that this very proximity can generate discomfort particularly in relation to stigmatized conditions like HIV, where fears of gossip and breached confidentiality undermine cooperation. Our findings extend the literature by showing that trust is not simply a product of interpersonal rapport but is co-constructed through institutional and systemic conditions such as perceived professional legitimacy (e.g., uniforms, ID badges, and visibility of government officials working with CHWs) and integration with the formal health system. This aligns with calls to view CHWs not as peripheral actors, but as part of health and governance systems [36]. In addition, the erosion of trust in cases where material or moral expectations went unmet highlights the contingent and negotiated nature of trust in everyday interactions, a dynamic that is often overlooked in technical accounts of CHW roles. These findings extend prior work on CHW-community relationships by showing the dual-edge of social proximity in rural settings [37–39]. Furthermore, by foregrounding the ambivalent effects of social proximity, this study contributes to a more nuanced understanding of the relational, institutional, and systemic terrains that CHWs must navigate, highlighting the need for training and systemic safeguards to enable CHWs to balance relational intimacy with professional distance, particularly in navigating sensitive health issues.

### 4.2. Strengthening governance and accountability

Our findings underscore how weak governance and accountability structures constrain CHWs' legitimacy, morale, and performance. Precarious employment, the absence of uniforms, inconsistent resource provision, and a lack of data feedback mechanisms collectively eroded CHWs' sense of professional identity and systemic belonging. These findings support Schneider's (2018) argument that CHW programme sustainability hinges on robust governance frameworks that institutionalize recognition, support, and accountability [20]. Moreover, we expand on Whidden et al., 2018 [40] by illustrating how operational feedback loops, particularly those involving routine reporting, performance monitoring, and data use, act not merely as managerial tools but as affective signals of inclusion and value. This study also adds to existing research that questions or challenges the view of CHWs being used mainly as tools or means to an end (i.e., to achieve short-term health system goals), rather than being treated as valued actors in the health system with their own agency, rights, and developmental needs [41]. Ultimately, these findings highlight an urgent need to integrate CHWs to formal health systems.

### 4.3. Reorienting the model of care

The shift from curative to community-based, preventative care has considerably expanded the scope of CHWs' responsibilities, yet systemic structures have struggled to keep pace with this transformation. In our study, CHWs were expected to fulfil both health-related and social support functions, ranging from medication delivery to assisting with grant applications, without commensurate institutional support or role clarification. These blurred expectations triggered mechanisms of overload, role conflict, and emotional fatigue, often resulting in diminished care quality or attrition. While existing literature recognises CHWs' intermediary role [42], our findings extend this by highlighting the conditions under which task expansion becomes counterproductive, even when community trust remains high. Specifically, we show that in the absence of engaged leadership, clear job descriptions, and workload management, CHWs are forced to negotiate competing demands with little formal institutional backing. This challenges policy narratives that frame task-shifting as a simple solution to workforce shortages and instead calls for a recalibration of CHW roles within integrated care models, to one that attends to fit, clarity, and institutional alignment rather than assuming infinite absorptive capacity.

### 4.4. Coordinating services within and across sectors

CHWs' ability to coordinate care across sectors is highly dependent on institutional responsiveness and coherent service integration. In contexts of fragmented governance, poor NGO-government collaboration, and persistent bureaucratic barriers, mechanisms of frustration, disempowerment, and erosion of public trust are activated. CHWs become default providers of social services, often unsupported and inadequately trained, reinforcing their marginalization. In contrast, when cross-sectoral collaboration is structured—through active war rooms, shared referral systems, and joint planning—CHWs can effectively link households to multisectoral support, enhancing care integration and reducing exclusion. This supports Afzal et al.,2021 [17] and extend their findings by foregrounding the relational and procedural mechanisms that either enable or constrain intersectoral collaboration. Furthermore, by foregrounding the institutional arrangements and everyday negotiations that either enable or constrain CHWs' coordinating roles, this study underscores that service integration is not merely technical, but deeply political and dependent on embedding CHWs within formal governance and decision-making structures.

### 4.5. Creating an enabling environment

The study highlights how an enabling environment is less shaped by formal policy and more by the lived reality of CHWs, including transport availability, peer learning systems, and emotional safety. In contexts of logistical shortages and stigma-ridden clinic settings, mechanisms of flexibility, moral commitment, and adaptive peer-learning are triggered, allowing CHWs to navigate gaps and provide responsive care. However, these adaptations come at a cost, leading to burnout, emotional fatigue, and inconsistent service quality. This extends the work of Johnson et al., 2022 who critique the celebratory framing of CHW "overperformance" as a result of a well-functioning system and dedication, but rather as a response from systemic neglect and failure [12]. Ultimately, this study emphasizes that enabling environments are not simply about resource provision, but also about creating institutional cultures that recognize, support, and protect CHWs' capacity to sustain high-quality care delivery over time.

The findings synthesized across the five IPCHS strategies reveal a deeply interconnected set of contexts and mechanisms that shape CHWs' capacity to deliver integrated people-centred health services. While each strategy offers a distinct lens, such as trust-building in engagement, institutional legitimacy in governance, or workload alignment in reorientation of the model of care, the findings of this study indicate that these are not discrete domains but mutually reinforcing components of a complex system. Meta-mechanisms such as trust, recognition, and motivation emerge repeatedly across strategies, functioning as cross-cutting enablers or constraints depending on the context. For example, trust not only facilitates disclosure at the household level but also mediates community perceptions of CHW legitimacy, engagement in care, and acceptance of referrals, spanning from engagement, governance, and service coordination.

In theory, the IPCHS framework captures the breadth of the CHW experience, but the empirical evidence here suggests that its strategies could be reconceptualized as a dynamic system rather than parallel pillars, where context-mechanism-outcome configurations cut across multiple domains. This points to a potential restructuring of the IPCHS application, not as five isolated strategies, but as an integrated, mechanism-sensitive model that centres relational, institutional, and structural coherence. Such a reframing would better reflect the complex realities of CHW programmes in underserved settings and enhance the practical use of the framework for implementation research and policy reform of CHW programmes.

## 4.6. Strengths and limitations

**4.6.1. Strengths.** The research implemented realist evaluation through IPCHS framework guidelines to examine CHW roles and functions while developing context-mechanism-outcome configurations to determine CHW success or failure in people-centered care delivery. The study combined artificial intelligence analysis tools with expert validation to identify patterns in CMO configurations which produced practical and strong theory-based results. Through the realist evaluation framework, middle-range theories were developed to explain which CHW interventions succeed for which people in which situations and through what processes for effective delivery people-centred care, which provided practical recommendations for policymakers and program managers working with limited resources.

A multidisciplinary research team consisting of public health, nursing, psychology, sociology and spatial epidemiology experts broadened and enhanced the analytical rigor of the study. The data collection process achieved high quality and cultural sensitivity through highly trained collectors who possessed master's qualifications and more than 30 years of combined experience in community-based research. Triangulation through structured observations, in-depth interviews, and focus group discussions (FGDs) further strengthened the study findings by incorporating diverse perspectives from CHWs, community members, and supervisors. This comprehensive approach enriched the interpretation of results, offering nuanced insights into CHW effectiveness.

In addition, this study makes significant methodological and empirical contributions to CHW research by combining realist evaluation, WHOs' IPCHS framework, AI-enhanced data analysis, and middle-range theory propositions.

**4.6.2. Limitations.** The research took place in five rural KwaZulu-Natal communities which provided detailed contextual insights but might not be applicable to urban settings or healthcare systems outside the region. The study team's on-site observation of CHW home visits potentially influenced the behavior of research participants despite efforts to reduce observer influence. The study results depended on self-reported information which might have introduced recall or social desirability biases. This study also did not collect demographic information of participants which would have provided more nuanced understanding of the results and added to describing context in realist evaluation.

Additionally, while thematic coding and AI-supported pattern recognition enhanced data interpretation, these approaches remain subject to researcher bias and interpretation. For example, the output generation of an AI tool is highly dependent on the strength and clarity of the input prompts. Also, even when using similar prompts, outputs may differ, because the AI tools can generate infinity number of answers to similar questions or queries, therefore reducing replicability of outputs. This is why the final team validation of AI generated outputs was critical in our analysis to mitigate these limitations and enhance the rigour of analysis. Finally, the exclusion of higher-level policymakers from the primary dataset limited our ability to explore systemic governance issues in full depth. These constraints should be considered when interpreting the findings and applying them to policy or practice.

## 4.7. Policy Implications

From a policy perspective, there is a critical need for the formal integration of CHWs into national healthcare systems. Governments should institutionalize CHWs as essential healthcare providers by providing permanent employment, fair wages, and professional development opportunities. This would improve job satisfaction, retention, and service quality. Additionally, clearly defined job roles and structured referral mechanisms should be established to prevent role conflict

and excessive workload, ensuring that CHWs remain focused on healthcare while facilitating social service linkages. Another key policy requirement is the standardization of resource distribution, ensuring that CHWs have consistent access to medical kits, gloves, uniforms, and documentation tools. This would enhance their credibility, motivation, and efficiency. Furthermore, developing a standardized salary structure with benefits is crucial to addressing financial insecurity among CHWs and reducing turnover. Multi-sectoral collaboration must also be reinforced by improving coordination between health services, social services, and other government institutions to optimize service integration and reduce the burden on CHWs. Finally, data transparency and feedback mechanisms should be strengthened by implementing real-time reporting systems that keep CHWs informed on how the data they collect is used to policy decisions.

## 5. Conclusion

This realist evaluation underscores that optimizing CHW roles for integrated people-centred care requires more than task reallocation, but requires structural reform, institutional accountability, and coordinated support for CHWs across systems. The interplay between context, mechanisms, and outcomes reveals that CHWs' performance is shaped by their embeddedness in community networks, professional recognition, and enabling environments. While CHWs demonstrate remarkable adaptability and commitment, sustainable impact of their services hinges on formalizing their roles, clarifying expectations, and ensuring systemic responsiveness. Furthermore, strengthening governance, cross-sectoral coordination, and supportive infrastructures is essential for transformation of the healthcare system to be people-centred. Future research should also include higher-level policymakers' perspectives as a primary focus because their viewpoints were not the main emphasis of this study to fully explore governance and policy implications.

## Supporting information

**S1 File. Case descriptions.**
(DOCX)

**S2 File. AI Prompts and example.**
(DOCX)

## Author contributions

**Conceptualization:** Usangiphile E. Buthelezi, André J van Rensburg, Mosa Moshabela, Inge Petersen.

**Data curation:** Usangiphile E. Buthelezi, Sanah Bucibo, Noxolisa Radebe.

**Formal analysis:** Usangiphile E. Buthelezi, André J van Rensburg.

**Funding acquisition:** André J van Rensburg, Zamasomi Luvuno, Arvin Bhana, Inge Petersen.

**Methodology:** Usangiphile E. Buthelezi, André J van Rensburg, Arvin Bhana.

**Project administration:** Tasneem Kathree.

**Resources:** André J van Rensburg, Inge Petersen.

**Supervision:** André J van Rensburg, Mosa Moshabela, Zamasomi Luvuno, Tasneem Kathree, Arvin Bhana, Inge Petersen.

**Validation:** André J van Rensburg, Mosa Moshabela, Arvin Bhana, Inge Petersen.

**Visualization:** Usangiphile E. Buthelezi, André J van Rensburg.

**Writing – original draft:** Usangiphile E. Buthelezi.

**Writing – review & editing:** Usangiphile E. Buthelezi, André J van Rensburg, Mosa Moshabela, Sanah Bucibo, Zamasomi Luvuno, Tasneem Kathree, Arvin Bhana, Inge Petersen.

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
