## [Decision Letter · Decision Letter 0]

10 Jun 2025

PGPH-D-25-01197

A Realist Perspective on Optimizing Community Health Workers’ roles and functions to deliver Integrated people-centred care

Dear Dr. buthelezi,

Thank you for submitting your manuscript to PLOS Global Public Health. After careful consideration, we feel that it has merit but does not fully meet PLOS Global Public Health’s publication criteria as it currently stands. Therefore, we invite you to submit a revised version of the manuscript that addresses the points raised during the review process.

We look forward to receiving your revised manuscript.

Kind regards,

Bipin Adhikari, MBBS, DTM&H, MCTM, MPH, DPhil

Academic Editor

Journal Requirements:

1. Please include a complete copy of PLOS’ questionnaire on inclusivity in global research in your revised manuscript. Our policy for research in this area aims to improve transparency in the reporting of research performed outside of researchers’ own country or community. The policy applies to researchers who have travelled to a different country to conduct research, research with Indigenous populations or their lands, and research on cultural artefacts. The questionnaire can also be requested at the journal’s discretion for any other submissions, even if these conditions are not met.  Please find more information on the policy and a link to download a blank copy of the questionnaire here: https://journals.plos.org/globalpublichealth/s/best-practices-in-research-reporting. Please upload a completed version of your questionnaire as Supporting Information when you resubmit your manuscript

2. We noticed that you used “unpublished data" in the manuscript. We do not allow these references, as the PLOS data access policy requires that all data be either published with the manuscript or made available in a publicly accessible database. Please amend the supplementary material to include the referenced data or remove the references.

Additional Editor Comments (if provided):

Reviewers' comments:

Reviewer's Responses to Questions

**Comments to the Author**

1. Does this manuscript meet PLOS Global Public Health’s publication criteria?

Reviewer #1: Yes

Reviewer #2: Yes

Reviewer #3: Yes

Reviewer #4: Yes

Reviewer #5: Yes

Reviewer #6: Yes

Reviewer #7: Yes

2. Has the statistical analysis been performed appropriately and rigorously?

Reviewer #1: Yes

Reviewer #2: Yes

Reviewer #3: Yes

Reviewer #4: Yes

Reviewer #5: Yes

Reviewer #6: Yes

Reviewer #7: N/A

3. Have the authors made all data underlying the findings in their manuscript fully available (please refer to the Data Availability Statement at the start of the manuscript PDF file)?

Reviewer #1: Yes

Reviewer #2: Yes

Reviewer #3: Yes

Reviewer #4: Yes

Reviewer #5: Yes

Reviewer #6: Yes

Reviewer #7: No

4. Is the manuscript presented in an intelligible fashion and written in standard English?

Reviewer #1: Yes

Reviewer #2: Yes

Reviewer #3: Yes

Reviewer #4: Yes

Reviewer #5: Yes

Reviewer #6: Yes

Reviewer #7: Yes

Reviewer #1: The presentation on findings could be made in more detail . Likewise, discussions could be aligned with the findings and with current advancements in the area. Besides that the manuscript is presented well.

Reviewer #2: Recommendations for improvement

1. Abstract: Mentioning of qualitative thematic coding and or qualitative CMO synthesis could stgrenthen the clarity for; clarify in the abstract that it is a qualitative study.

2. Discussion: Consider adding a paragraph on at the end to clearly summarise study constraints such as sampling scope,generalisability, and limitation of analysis techniques, etc.

3. In the final submission, ensure figures (Fig 1–7) are fully integrated and properly captioned; they are crucial to illustrating CMO configurations and program theories.

Reviewer #3: Data Availability: Update the Data Availability Statement to reflect that the data will be fully and openly accessible on Figshare upon publication, providing a direct link or accession number. Clarify that no separate request to the supervisory team will be required for access if the data is indeed made publicly available.

AI Methodology Details: Consider adding a supplementary appendix or expanding the methods section to include specific examples of the prompts used for DeepSeek AI and a more detailed explanation of how the "R1 reason feature" was employed and its outputs interpreted and validated. This would greatly enhance transparency and reproducibility.

Policymaker Perspective: While acknowledged as a limitation, the discussion and conclusion could further emphasize how the identified governance challenges and proposed middle-range theories directly inform and guide policy reforms, even without direct input from high-level policymakers in this particular study.

Reviewer #4: Thank you so much for the opportunity to review this manuscript. The research scope and goals are valuable to improving the success of the utilization of CHWs, especially in developing nations and rural areas where this research was conducted. Having adequately reviewed this manuscript, it aligns with the publication criteria for this journal. The study is methodologically rigorous and technically sound. The methodology adopted was clearly explained, comprehensive, and based on the goals of the research, and it was also very suitable and justified. The development of middle-range theories further justifies the practicality of the study and highlights their contribution to existing literature. The analytical tools and processes adopted for the qualitative data were also very clear and rigorous. It was helpful that a diagram was also attached for the data analysis. The authors have also declared their willingness to make the data available upon request, which shows transparency and aligns with the journal’s requirement on data. The manuscript was largely easy to read and generally well-written in standard English. However, there are areas where conciseness could be improved. For instance, in the discussion section, revisiting previously explained mechanisms without adding new interpretation may lengthen the manuscript unnecessarily. For instance, restating the issues of CHW motivation and trust multiple times could be streamlined. This was observed throughout the manuscript. Avoid unnecessary repetition to reduce redundancy. The use of figures to explain and identify the relationship between the concepts, mechanisms, and outcomes was very thoughtful and useful in guiding the readers considering the density of the manuscript.

To improve this manuscript

1. It requires more discussion on how the findings align or contradict existing literature, which I believe was not strongly established in the discussions section. You mentioned it briefly but not in all sections and it may also be beneficial to have a more detailed discussion to highlight the significance of this study.

2. Adopting middle-range theories was an important contribution of this study as it adds depth to the understanding of systemic barriers and facilitators affecting CHW roles. Consider offering more concrete policy implications that stakeholders could implement based on your findings.

3. The use of artificial intelligence in data analysis is recently evolving; you should briefly discuss how this may influence bias in data analysis or results due to AI bias.

4. The presentation and labeling of the tables could be improved. It was sometimes difficult to identify which concept was associated with what outcome.

Overall, this was a very beautifully written manuscript. You just need minor revisions to make it better.

Reviewer #5: The manuscript is concise and details it's findings in a presentable manner. It presents evidence useful in the relevant area and is recommended for publication in the journal at present. Ethical concerns have been met.

Reviewer #6: The authors offer a thoughtful and policy-relevant findings drawn from a realist evaluation of community health worker (CHW) roles within integrated, people-centred care systems. It successfully moves beyond a narrow focus on task reallocation to highlight the deeper structural, institutional, and systemic transformations required to optimize CHW contributions. This reflects a clear understanding of the realist evaluation framework, particularly the importance of examining how context and mechanisms interact to shape outcomes.

Reviewer #7: Title: The title reflects the focus of the article.

Abstract: Is reflective of the content of the article.

Introduction

The introduction section is well-structured and concise while providing sufficient information to offer a comprehensive overview.

The authors use numerous abbreviations, which may become confusing for the reader, as they need to repeatedly refer back to their definitions. Consider reducing the use of abbreviations that appear infrequently by writing them out in full, such as OTL and OM.

The problem is clearly articulated.

Methodology

The researchers should be commended for employing a realistic evaluation approach, as it provides a novel perspective on a phenomenon that has already been explored in the literature.

Overall, the methodology is reported in detail; however, some sections lack clarity. For example, in Step 1, the practical application is not clearly outlined. In Step 2, it is unclear what information was gathered during each data-gathering activity. Given the multiple data collection methods (interviews, observations, and focus groups), it is difficult to determine which data was collected at each stage. Additionally, the process of recording and analysing observations is not clearly explained.

The data-capturing methods and opportunities are extensive and complex. Consider summarising this information in a diagramme or table that provides a visual overview of the data sources, data collection tools/methods, and the focus of each data-gathering activity (i.e., what each interview, observation, or focus group sought to collect in response to the research question). This could enhance the accessibility of the methodology for a broader audience.

It is also unclear whether the coding approach was inductive or deductive. See specific comments related to other steps.

It is also unclear whether the coding approach was inductive or deductive. See specific comments related to other steps.

Results

Why was no demographic information included?

Overall, the results are comprehensive and detailed. However, the volume of information presented may be overwhelming, making it difficult to track all findings. It might be beneficial to divide the results into two separate articles. Please refer to the manuscript for specific comments on this section.

Discussion

The discussion is comprehensive and includes appropriate detail.

References

Please update all references to correctly reflect the use of "et al."

Future Research

It would also have been valuable to compare governance, access to resources, and other factors across the five different communities to examine their influence on role performance.

**Do you want your identity to be public for this peer review?** For information about this choice, including consent withdrawal, please see our Privacy Policy

Reviewer #1: **Yes: ** Jemish Acharya

Reviewer #2: No

Reviewer #3: **Yes: ** Abdulmalik Alilu Abubakar

Reviewer #4: No

Reviewer #5: **Yes: ** Jemish Acharya

Reviewer #6: No

Reviewer #7: No

---

## [Decision Letter · Decision Letter 1]

20 Aug 2025

A Realist Perspective on Optimizing Community Health Workers’ roles and functions to deliver Integrated people-centred care

PGPH-D-25-01197R1

Dear Mr buthelezi,

We are pleased to inform you that your manuscript 'A Realist Perspective on Optimizing Community Health Workers’ roles and functions to deliver Integrated people-centred care' has been provisionally accepted for publication in PLOS Global Public Health.

Best regards,

Bipin Adhikari, MBBS, DTM&H, MCTM, MPH, DPhil

Academic Editor

Reviewer Comments (if any, and for reference):

Reviewer's Responses to Questions

**Comments to the Author**

Reviewer #2: All comments have been addressed

Reviewer #3: All comments have been addressed

publication criteria?

Reviewer #2: Yes

Reviewer #3: Yes

3. Has the statistical analysis been performed appropriately and rigorously?

Reviewer #2: Yes

Reviewer #3: Yes

4. Have the authors made all data underlying the findings in their manuscript fully available (please refer to the Data Availability Statement at the start of the manuscript PDF file)?

Reviewer #2: Yes

Reviewer #3: Yes

5. Is the manuscript presented in an intelligible fashion and written in standard English?

Reviewer #2: Yes

Reviewer #3: Yes

Reviewer #2: All comments raised during the first review have been satisfactorily addressed.

Reviewer #3: Based on the thorough review of the provided manuscript, which includes the authors' responses to prior feedback, the article is of high quality. The study employs a rigorous realist evaluation approach to address a critical knowledge gap in the public health sector. The authors have been transparent and responsive in their revisions, addressing all editor and reviewer comments appropriately. The methodology is sound, the findings are significant, and the conclusions provide valuable policy recommendations.

**Do you want your identity to be public for this peer review?** For information about this choice, including consent withdrawal, please see our Privacy Policy

Reviewer #2: No

Reviewer #3: No
